# Effects of Shelterbelt Transformation on Soil Aggregates Characterization and Erodibility in China Black Soil Farmland

Tongwei Kong [1], Binhui Liu [1,*], Mark Henderson [2], Wanying Zhou [1], Yuanhang Su [1], Shuai Wang [1], Ligang Wang [3,4] and Guibin Wang [1]

1 College of Forestry, The Northeast Forestry University, Harbin 150040, China
2 Mills College, Northeastern University, Oakland, CA 94613, USA
3 Qiqihar Branch of Heilongjiang Academy of Forestry, Qiqihar 161005, China
4 National Permanent Scientific Research Base on Shelter Forest in Nenjiang Plain, Qiqihar 161005, China
* Correspondence: binhui@nefu.edu.cn; Tel.: +1-3100881689

**Abstract:** Farmland shelterbelts are widely used to reduce wind erosion, maintaining the ecological and food security of the black soil plain region of northeast China. In recent years, the protective effect of shelterbelts has been reduced due to tree degeneration. Efforts have been made to transform the construction of shelterbelts to conserve the stability of soil aggregates and enhance protection against erosion, however, the results are not well understood. To evaluate the impact of shelterbelt transformation on the stability of farmland soil aggregates and soil erodibility, three transformation modes of shelterbelts were selected, including pure *Pinus sylvestris* var. *mongolica* (ZC), pure *Picea asperata* (YS), and mixed *Populus × xiaohei–Pinus sylvestris var.mongolica* (ZY), with a degraded *Populus × xiaohei* shelterbelt (TYC) used as a control. We set up soil sampling points at 0.5H, 1H, 1.5H, 3H, 5H, 7H, and 9H from the shelterbelts and analyzed aggregate composition, mean weight diameter (MWD), geometric mean diameter (GMD), fractal dimension (D), soil erodibility (K-value), and their relationships to soil properties of the 0–10 cm, 10–20 cm and 20–40 cm soil layers and the shelterbelt structure by using dry and wet sieving and equation estimation methods. The results show that dry (d) sieved soil samples from the transformed shelterbelt-protected farmlands are mainly composed of 2–5 mm and >5 mm grain size aggregates; the sum of the two particle sizes ranged from 48.67% to 51.27%, significantly larger than in the degraded shelterbelts (15.37%), decreasing with increasing distance from the shelterbelts. The effect is most obvious in the 0–10 cm soil layer. Wet (w) sieved soil samples are all dominated by <0.25 mm and 0.25–0.5 mm grain size aggregates; the sum of the two particle sizes ranged from 78.25% to 80.82%, which do not vary significantly with the mode of shelterbelts. The dMWD and dGMD show significantly higher mean values in samples from transformed shelterbelt-protected farmland than in soil from degraded shelterbelt-protected farmland; their magnitudes differ depending on the transformation mode, showing a pattern of ZC > ZY > YS and decreasing with increasing distance from shelterbelts, while the opposite is true for D and K. The difference between wMWD and wGMD for different shelterbelts protected farmland is not significant and is significantly lower than that between dMWD and dGMD. Clay and silt content was highly significantly positively correlated with aggregates dMWD and dGMD, weakly positively correlated with wMWD, wGMD and wD, and highly significantly negatively correlated with dD and K values. This shows that particle composition parameters can be used to reflect the sensitivity of agricultural soils to wind erosion. Farmland shelterbelt porosity is the main factor driving changes in soil aggregates stability, soil erodibility, and other soil properties. The transformation of degraded farmland shelterbelts can decrease the porosity and reduce wind speed, resulting in improved stability and erosion resistance of the farmland soil aggregates by increasing the clay content of the farmland soils. These results are useful in renovating degraded shelterbelts, providing novel insights into how to regulate the stability of soil aggregates and soil erodibility characteristics at the shelterbelt network scale.

**Keywords:** aggregate stability; geometric mean diameter; K value; mean weight diameter; soil erodibility

## 1. Introduction

Soil aggregates are a vital indicator of soil structure and soil quality [1]: their quantity and stability play an important role in resisting erosion and supporting vegetation growth [2]. Measurements of soil aggregate stability and particle size composition allow for the indirect comparison of the erosion resistance of different soils [3]. In addition, the stability of soil aggregates affects a wide range of physical and biogeochemical processes in natural and agro-ecosystems [4]. Thus, sound soil structure and high aggregate stability are important factors in influencing soil fertility, maintaining soil productivity, and reducing soil erodibility [5]. Indicators such as mean weight diameter, geometric mean diameter, and fractal dimension have been widely used for the quantitative evaluation of soil structural stability [6]; aggregates of different sizes play complementary roles in maintaining soil stability and improving soil structure [7]. Excellent aggregate structure significantly improves the physical protection of organic carbon and is important for enhancing soil fertility and sustainable soil health [8].

The black soil region in northeast China, one of the four main global regions of black soil, is distributed across Heilongjiang, Jilin, and Liaoning provinces and the Inner Mongolia autonomous region, with a total area of approximately $103 \times 104 \text{ km}^2$ [9]. It is the soil with the highest organic matter content in China's agricultural soils. The region has been heavily cleared for agriculture in just the past century, with a reclamation rate of over 70% [10]. Northeast China's cropland area and grain production each account for a quarter of the country's total, making it an important commercial grain base, known as the "stabilizer" of China's food security [11]. Though the black soils of northeast China are rich in soil organic matter and have a high fertility compared to other soils [12], long-term overcultivation combined with serious wind erosion have led to a dramatic decline in the quality and quantity of the black soil [13]: about half of the soil N and organic matter has been lost from the black soils of northeastern China [14]. In particular, increasing wind erosion in the western plain of Heilongjiang is a serious threat to the ecological environment and the grain production capacity of the black soil region [15].

Research has shown that planting trees to form shelterbelts around farmland can be effective in arid and semi-arid areas. Functioning to reduce wind speeds, control soil erosion, and improve microclimates [16–18], shelterbelts can be an effective measure to stabilize crop production while also promoting ecological landscape functions [19]. At the same time, the construction of shelterbelts causes changes in the structure and fertility of agricultural soils [20,21], which are important to protect farmland ecosystems and maintain the agricultural ecological balance. For instance, studies have shown that shelterbelts on the plains can effectively reduce wind erosion by 22–60% and can also regulate the fertility and the physical and chemical properties of the soil [21].

The construction of farmland shelterbelts in the black soil region of western Heilongjiang province dates back to the 1950s and accelerated as part of China's "Three Norths" shelterbelt system project since 1978. The network of shelterbelts protecting farmland in this region is now quite extensive but is not without some negative effects. During the early construction of farmland shelterbelts, planting shelterbelts consisting of pure *Populus* (a species with short growth cycle) was the norm, and the number and scale of shelterbelts were emphasized, neglecting the relationship between the structural configuration of the shelterbelts and its protective benefits. Planting shelterbelts that relied on a single tree species, especially one with short growth cycle, reduced the functional lifespan of the shelterbelts' effects in controlling wind erosion, which in turn led to the degradation of farmland soils.

Parts of the black soil region in western Heilongjiang province have experienced sudden declines in the protective benefits of the shelterbelts for arable land and serious degradation of the quality and quantity of black soil [22,23]. Finding new measures to improve ecological conditions is important for restoring agricultural production. In other countries with black soil regions, no-till and minimum-till agro-pastoral practices are effective conservation measures to reduce erosion and protect farmland [24], though in

the short term they tend to produce lower yields. In China, due to concerns about the small amount of arable land per capita, it is difficult to achieve no-till and low-till practices. Instead, attention has been focused on transforming the configuration of shelterbelts to improve their protective effects, reducing erosion and increasing the structural stability and productivity of agricultural soils [25]. Research to better understand the effects of different farmland protection shelterbelts on farmland soil structure is important to the conservation of black soil resources in northeast China.

There is an abundance of prior research on farmland protection shelterbelts [26–28]. Studies have shown that farmland shelterbelts can improve the ecological environment, enhance the environmental conditions required for crop growth, and have a significant positive impact on the growing environment of crops [29–31]. To date, there are more studies on wind and sand control and microclimate improvements from farmland shelterbelts in China [32–34], but fewer studies on soil improvement and agroecosystem improvement; the effects of farmland shelterbelts on soil aggregate characteristics and soil erodibility have not previously been reported. This paper assesses the effects of transformation of farmland shelterbelts on the aggregate stability and erodibility of farmland in the black soil area of the semi-arid plains, analyzing their relationship to the shelterbelt structural characteristics and soil properties. The aim is to provide a theoretical basis for the restoration and transformation of degraded shelterbelts to improve their effectiveness in protecting farmland, with a view to maximizing their ecological and economic benefits and their sustainable development. This study applies both wet and dry sieving methods to characterize soil aggregates and the soil erodibility of farmland after the renewal of farmland protection shelterbelts in the black soil area of Northeast China. The objectives of the study were, first, to assess and compare the effects of different shelterbelt transformation methods on soil aggregates and soil erodibility, and second, to explore the relationship between soil aggregate characteristics and soil erodibility with soil physicochemical properties and the shelterbelts' structural characteristics.

## 2. Materials and Methods

### 2.1. Study Area

The study site is located in Gannan County, in the northwestern part of the Songnen Plain (47°44′–48°40′ N, 123°36′–123°48′ E; see Figure 1). The area, typical of the black soil region, is arid to semi-arid, with a dry and cold high latitude continental monsoon climate: During 1987–2017, the average annual total precipitation was 438.1 mm, mainly concentrated in summer months. The average monthly high temperature in the hottest month (July) was 23.3 °C and the average monthly low temperature in the coldest month (January) was −18.3 °C (Figure 2). The average annual evaporation is 1199.6 mm. The frost-free period (90% frequency) is 125 d, with a soil freezing period from early November to mid-May and a maximum freezing depth up to 2.5 m. The landscape marks the transition into the alluvial plains of the Arun River and Yin River basins, with altitudes ranging from 160 to 190 m. The soil type is mainly black calcareous, developed from wind and sand deposits. The texture is sandy loam and alluvial sand with a high sand content; the soil depth is 30–40 cm, with a pH of 6.5–7.5. Frequent high winds in winter and spring have led to wind erosion disasters and destructive sand deposition.

In order to protect the farmland from wind erosion in the spring and water erosion in the summer rainy season, the region began to establish a network of mainly poplar (*Populus × xiaohei*) based farmland shelterbelts in the late 1970s. In recent years, in response to the widespread degradation of poplar shelterbelts, some degraded shelterbelts have been transformed by replacing poplar with pine, the main species being *Pinus sylvestris var. mongolica* and *Picea asperata*.

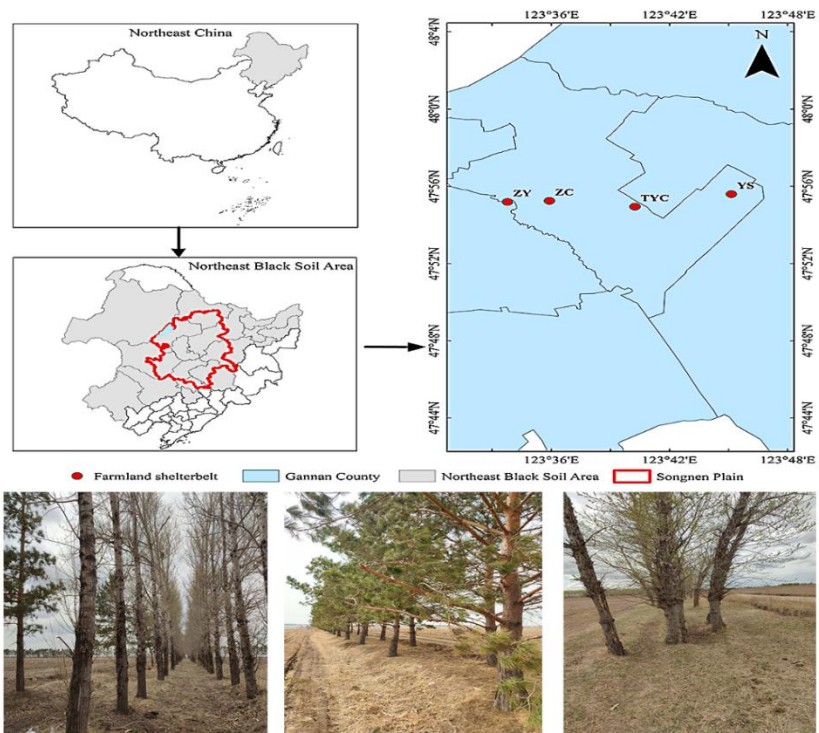

**Figure 1.** Research region and sample site.

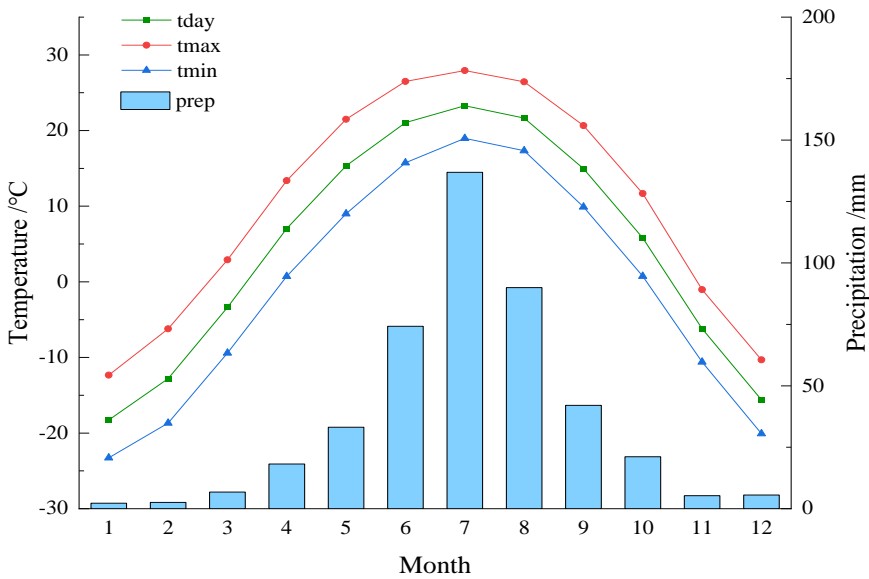

**Figure 2.** General climatological conditions in the study region.

## 2.2. Research Methods

### 2.2.1. Soil Sampling

Shelterbelts were selected for this study through a field survey before the spring ploughing in 2021. Each sample site represented one of the three main shelterbelt transformation modes—pure *Pinus sylvestris var. mongolica* (ZC), pure *Plicae asperata* (YS), or mixed *Populus* × *xiaohei–Pinus sylvestris var. mongolica* (ZY)—with the untransformed, degraded *Populus* × *xiaohei* shelterbelts (TYC) is used as a control.

In selecting these sites, we favored well-preserved shelterbelts with good spatial homogeneity of soil structure, remote from man-made structures, and consistent alignment of the trees making up the shelterbelt. All of the transformed shelterbelts had been planted

following the clearcutting of degraded Populus shelterbelts in 1991–1993 and have the same soil parent material and management history. The shelterbelts are the same in terms of basic stand conditions and management patterns. Within each study shelterbelt, we set up three standard sample plots of 10 m × 6 m (length × width) and measured tree height, diameter at breast height, tree age, shelterbelt permeability, tree spacing within rows, and spacing between rows. The basic information on the farmland shelterbelts is shown in Table 1.

**Table 1.** Basic situation of farmland protection shelterbelt.

| Shelterbelt Species Composition | Shelter-Belt Abbreviations | Average Tree Age (Years) | Average Tree Height/(m) | Mean DBH (cm) | Porosity (%) | Plant Interval and Row Interval/(m) |
|---|---|---|---|---|---|---|
| *Populus × xiaohei-Pinus sylvestris var. mongolica* | ZY | 29 | 10.40 | 24.20 | 0.46 | 2.5 × 4.5 |
| *Pinus sylvestris var. mongolica* | ZC | 28 | 10.63 | 27.07 | 0.36 | 3.0 × 2.0 |
| *Piceaasperata* | YS | 28 | 7.27 | 23.25 | 0.43 | 3.5 × 2.0 |
| *Populus × xiaohei* (degraded) | TYC | 45 | 9.70 | 27.71 | 0.74 | 3.0 × 1.5 |

After selecting the study shelterbelt sites, we determined the location of the soil sampling points in terms of the height multiplier of the average tree height (H). Three sample lines were drawn in parallel within 50 m on each side of the mid-plumb line of the selected main stand of each shelterbelt, and each sample line was delineated with sampling points measured from the main belt at 0.5H (the edge of belt), 1H, 1.5H, 3H, 5H, 7H, and 9H. The sample points are shown in Figure 3. Information about the amount of fertilizer applied and other farmland management measures at each site was provided by the local agricultural department and verified in the field. To ensure that the soil samples collected were representative of both the agricultural land and the selected shelterbelt, we removed dead leaves and other impurities such as crop residues from the soil surface before digging into the soil profile. The soil bulk density was measured at 0–10, 10–20 and 20–40 cm using a 100 cm$^3$ circle cutter. At each layer, we collected 1.5 kg of soil in its original state in a plastic lunch box, with a total of 252 soil samples taken back to the laboratory. There, we extracted 3/4 of each sample and, gently breaking up the larger soil clumps in the soil sample into small clumps of about 1 cm$^3$ along their natural cracks, we removed small stones, plant residues and other organic materials from the sample. We then placed the soil in a cool and ventilated location to dry naturally for grading the soil aggregates, air-drying the remaining soil samples.

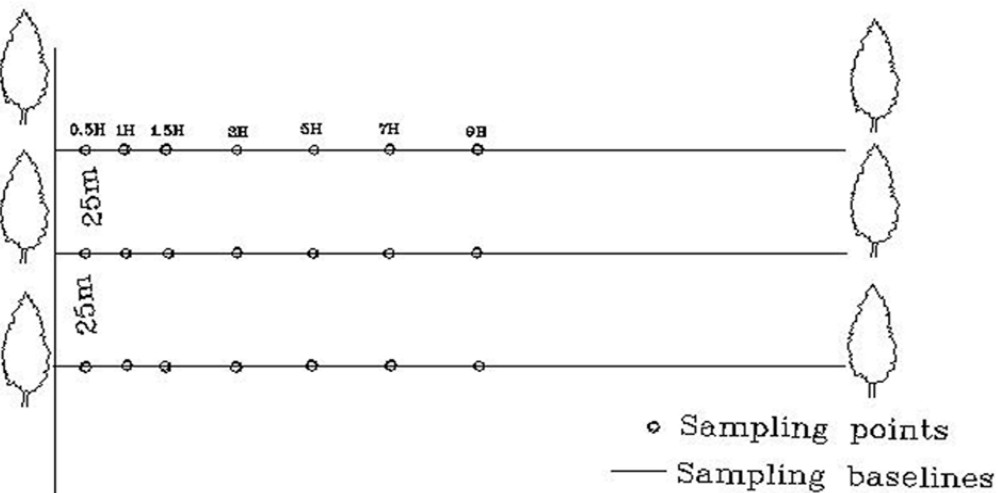

**Figure 3.** Sample point setup diagram.

2.2.2. Sample Indicator Determination and Methods

We followed the methods reported by Zhang [35] in measuring various soil indicators: soil bulk density was determined using the circular cutter method [36], soil organic carbon (SOC) was determined using the dichromate oxidation method [37], the Kjeldahl method was used to analyze total soil nitrogen (TN) [38], and soil mechanical composition was determined using the pipette method [35]. We applied both wet and dry sieving methods [39]. After soil samples of each layer were naturally dried, a 200 g sample was weighed through sieves with pore sizes of 5, 2, 1, 0.5 and 0.25 mm, and the weight of each particle size was measured separately. We then obtained particle size aggregates of >5 mm, 2–5 mm, 1–2 mm, 0.5–1 mm, 0.25–0.5 mm, and <0.25 mm. The wet sieve was performed by making a 50 g soil sample with proportional amounts of each particle size in the dry sieve, which was then placed in the aggregate analyzer; this we first moistened slowly with water for 5 min, then set the frequency to 30 cycles/min and the amplitude to 3 cm for 10 min. After washing the soil particles on each sieve surface into a weighed aluminum box, they were dried at 60 °C to a constant weight. We measured the weights for each particle size (accurate to 0.01 g) and calculated the mass fraction of each particle size, obtaining the soil aggregates of >5 mm, 2–5 mm, 1–2 mm, 0.5–1 mm, 0.25–0.5 mm, and <0.25 mm.

Soil aggregate stability indicators *MWD* and *GMD* are calculated using the following equations [39]:

$$MWD = \sum_{i=1}^{n} \overline{R}_i W_i \tag{1}$$

$$GMD = exp\left[\frac{\sum_{i=1}^{n} W_i ln\overline{R}_i}{\sum_{i=1}^{n} W_i}\right] \tag{2}$$

where *MWD* is the mean weight diameter of the soil, mm; *GMD* is the geometric mean diameter, mm; $\overline{R}_i$ is the mean diameter of a grain class, mm; and $W_i$ is the mass percentage of aggregates in grain size *i* %. The fractal dimension *D* was calculated with reference to the derivation of Yang, et al. [40]:

$$\frac{M_{(r<\overline{R}_i)}}{M_T} = \left[\frac{\overline{R}_i}{\overline{R}_{max}}\right]^{3-D} \tag{3}$$

where *D* is the fractal dimension of the soil aggregates; $M_{(r<\overline{R}i)}$ is the cumulative mass of soil particles with a particle size smaller than $\overline{R}_i$, g; and $\overline{R}_{max}$ is the average diameter of the aggregates with the largest particle size, mm.

Soil erodibility K values are an important parameter in soil erosion forecasting. The equations developed by Shirazi show a good linear relationship between the erodibility values calculated and the measured values. Following Shiriza and Boersma [41], Equation (4) was used to calculate soil erodibility K values:

$$K = 7.954 \times \left\{0.0017 + 0.0494 \times exp\left[-0.5 \times \left(\frac{\log GMD + 1.675}{0.6986}\right)^2\right]\right\} \tag{4}$$

*2.3. Statistical Analysis*

The data were statistically analyzed using Microsoft Excel 2019 and IBM SPSS Statistics 26.0 software. The differences in physicochemical properties, wet and dry aggregate characteristics and erodibility K-values of the soils were analyzed by one-way ANOVA. The correlation between soil dry and wet aggregate stability characteristics and K values and soil properties was explored by Pearson's correlation test, with a significance level of 0.05. Using Canoco 5.0 software, redundancy analysis (RDA) was used to determine the relative contribution of stand thinning, mean shelterbelt height, distance from shelterbelt, soil texture, and soil properties in explaining differences in soil stability and soil erodibility in different soil layer variables. The PCA method uses factor analysis to determine the eigenvalues and eigenvectors of the principal components of the indicators, selecting the

key principal components based on cumulative contribution, calculating the scores of each principal component, and then using the composite score equation to compute the F value. In this study, the composite score for soil aggregate stability was derived by using a combined principal component analysis (PCA). Soil physical and chemical properties (soil bulk density, soil organic matter, soil total nitrogen, and soil mechanical composition), as well as aggregate particle size distribution and indicators of aggregate stability such as dMWD, dGMD, dD, wMWD, wGMD, wD values and K values were selected as initial variables. Based on the principle of eigenvalues greater than 1, the common factors *F1*, *F2*, *F3* and *F4* were selected using factor analysis. On this basis, the weights of *F1*, *F2*, *F3* and *F4* were used to calculate the total score (*F*) [42]:

$$F = F1\frac{36.386}{79.031} + F2\frac{18.468}{79.031} + F3\frac{13.247}{79.031} + F4\frac{10.930}{79.031} \tag{5}$$

where 36.386 is the contribution of factors selected on co-factor $F_1$; 18.468 is the contribution of factors selected on co-factor $F_2$; 13.247 is the contribution of factors selected on co-factor $F_3$; 10.930 is the contribution of factors selected on co-factor $F_4$; and 79.031 is the contribution of factors selected on co-factors $F_1$, $F_2$, $F_3$ and $F_4$ sum. Ultimately, we calculated the total score *F* for the stability of soil aggregates on agricultural land at different distances from shelterbelt.

## 3. Results

### 3.1. Effect of Shelterbelts Transformation on Farmland Soil Properties

The change in soil bulk density of different soil layers in each shelterbelt's protected farmland is shown in Figure 4a. The distance from shelterbelts has a certain influence on the soil bulk density: the lowest soil bulk density is found from 0.5 to 1H, generally increasing with distance from the shelterbelts, and is most evident in the surface layer (0–10 cm). As soil depth increases, soil bulk density shows an overall increasing trend. The soil bulk density of farmland in transformed shelterbelts (YS, ZC, ZY) is significantly lower than that of degraded forest belts (TYC), and there are differences among the transformation methods. This shows that the shelterbelts are associated with higher levels of soil bulk density in farmland, especially in the surface layer, but the effect declines with increasing distance from the shelterbelts.

The variation of soil organic matter and total nitrogen in different soil layers in each shelterbelt protected farmland is shown in Figure 4b,c. Overall, the organic matter and total nitrogen content of transformed shelterbelts (YS, ZY, ZC) are greater than those of degraded shelterbelts (TYC). These measurements vary with increased distance from the shelterbelts: the soil organic matter content in the YS and ZY shelterbelts showed a "W" pattern, followed by a "V" pattern in the ZC shelterbelt and an "M" pattern in the TYC shelterbelt. In terms of vertical variation, the organic matter and total nitrogen content of farmland soils in each shelterbelt tends to decrease with increasing soil depth, and the organic matter content of the 0–10 and 10–20 cm soil layers is more influenced by the distance from the shelterbelts than the 20–40 cm soil layer.

The mechanical composition of the soil in each shelterbelt is shown in Figure 4d–f. Overall, sand content is significantly higher than clay and powder content across shelterbelt types and depths. There are significant differences in the mechanical composition of the soil in the protection range of each shelterbelt. The mean value of the clay content in each soil layer is larger in the transformed shelterbelts (YS, ZC, ZY) than in the degraded shelterbelt (TYC), and there are also differences among different transformation methods, with ZC > ZY > YS. In particular, the effect on the clay content of the surface soil is significant, and the difference in clay content between shelterbelts decreases as the depth of the soil layer increases. By contrast, sand content shows the opposite pattern, with YS > ZY > ZC.

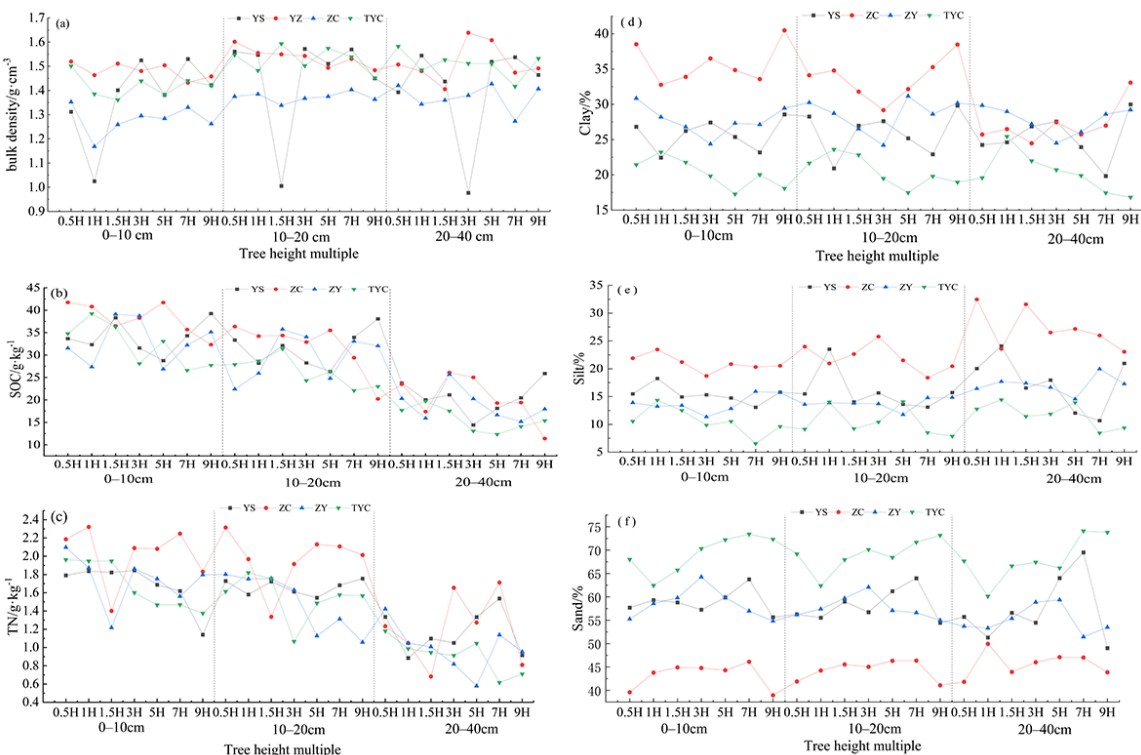

**Figure 4.** Changes in soil properties in the 0–10, 10–20 and 20–40 cm soil layers. Notes: (**a**) means Bulk density; (**b**) means SOC; (**c**) means TN; (**d**) means Clay content; (**e**) means Silt content; (**f**) means Sand content.

### 3.2. Effect of Shelterbelts Transformation on the Characteristics of Soil Aggregates

#### 3.2.1. Soil Aggregate Grain Size Distribution

The distribution of soil dry grain sizes in different shelterbelt-protected farmlands are shown in Figure 5. Overall, the transformed shelterbelts (YS, ZC, ZY) are mostly made up of >5 mm and 2–5 mm size aggregates, with the two particle sizes together accounting for 48.67% to 51.27% of the total. In the degraded shelterbelt (TYC), 0.25–0.5 mm and <0.25 mm size aggregates predominate, together accounting for 64.63% of the total. Different characteristics with increasing distance from the shelterbelts were observed among different types of shelterbelts, with YS showing a trend of increasing and then decreasing grain size aggregates in the 0–10, 10–20 and 20–40 cm soil layers (>5 mm and 2–5 mm) with increasing distance, and ZC, ZY and TYC showing a trend of decreasing grain size aggregates with increasing distances, all with a significant decrease at 7H, yet the 0.25–0.5 mm and <0.25 mm grain size aggregates showed the opposite trend. Among these, the 1–2 mm and 0.5–1 mm grain size aggregates are relatively stable among the different shelterbelt types. The shelterbelt transformation mode influences the grain size distribution of aggregates, with large aggregates (>0.25 mm) showing ZC > ZY > YS; >5 mm and 2–5 mm grain size aggregates gradually increasing with soil depth; and 0.25–0.5 and <0.25 mm grain size aggregates gradually decreasing with soil depth.

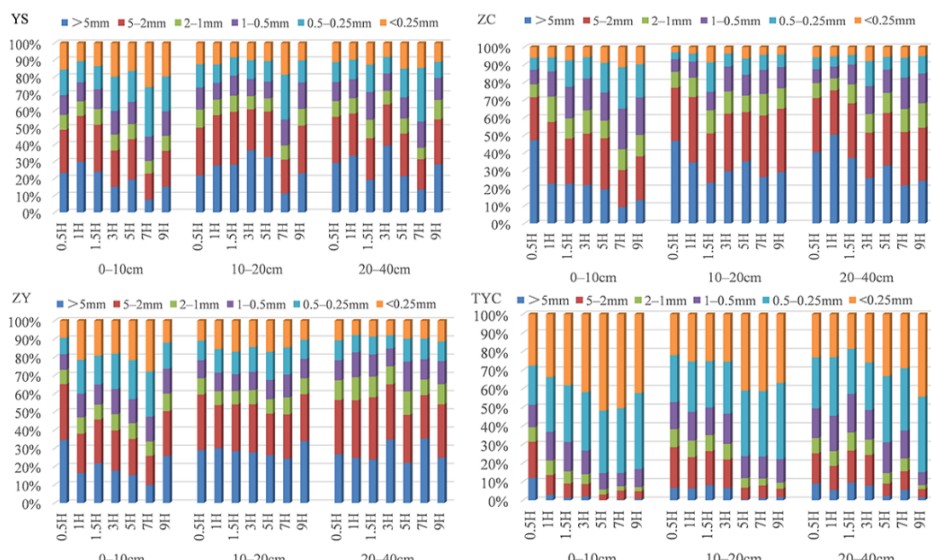

**Figure 5.** Soil dry stable aggregates distribution.

The distribution of wet sieve soil grain sizes in different shelterbelt-protected farmlands are shown in Figure 6. Overall, 0.25–0.5 mm and <0.25 mm grain size aggregates make up the majority of the aggregates; the sum of the two particle sizes ranged from 78.25% to 80.82%, while >5 mm and 2–5 mm grain size aggregates make up a decreasing proportion. In the 0–10, 10–20 and 20–40 cm soil layers, the distance from the shelterbelt only had an effect on the grain size of the <0.25 mm water-stable aggregates, with a fluctuating pattern of variation with increasing distance, while the effect on the 1–2 mm, 0.5–1.0 mm and 0.25–0.5 mm grain size aggregates was not significant.

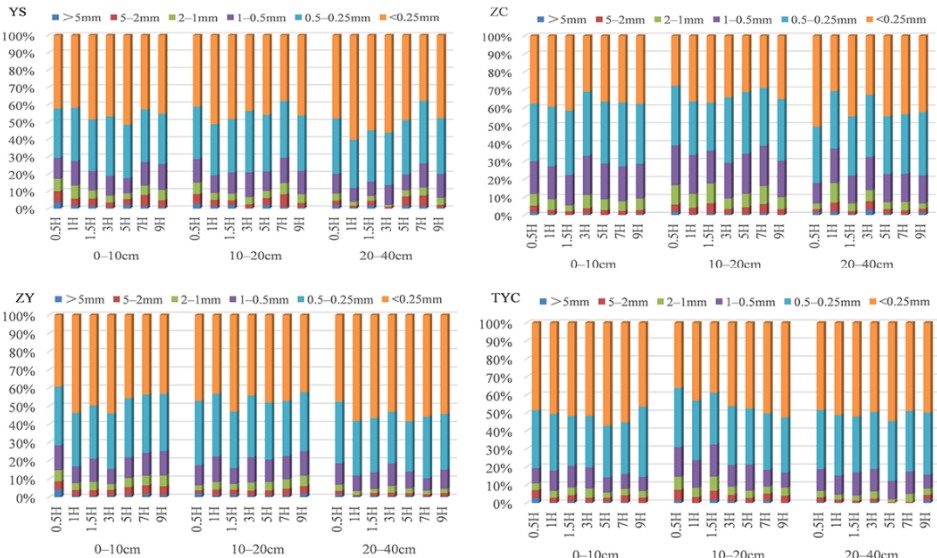

**Figure 6.** Soil water stable aggregates distribution.

Examining the dry and wet sieve agglomerate size distribution, we found that, compared with the degraded shelterbelts, the transformed farmland shelterbelts can significantly change the soil dry aggregate size distribution, increasing aggregate sizes from small to large. Different shelterbelt transformation methods achieved different results. Distance from the shelterbelts also has an effect on the grain size distribution of aggregates. The closer to the shelterbelt, the higher the content of large aggregates (>0.25 mm), though the

values fluctuate and we found a significantly higher effect on the grain size distribution of aggregates in dry sieves than in wet sieves.

### 3.2.2. Soil Aggregate Stability (dMWD, dGMD, dD, wMWD, wGMD and wD)

The MWD and GMD of dry and wet soil aggregates at different shelterbelt-protected farmlands are shown in Figure 7. Overall, dMWD and dGMD showed significantly larger mean values in transformed shelterbelts (YS, ZC, and ZY) than in degraded forest belts (TYC), and the transformation methods differed on their sizes, showing ZC > ZY > YS, while wMWD and wGMD sizes did not differ significantly between shelterbelts. In the 0–40 cm soil layer of YS, ZC, ZY and TYC shelterbelts, the mean dMWD values were 2.96, 3.50, 3.01 and 1.07 mm, and the mean dGMD values were 1.55, 2.24, 1.70 and 0.51 mm, respectively. The dMWD and dGWD showed a decreasing pattern with the increased distance from the shelterbelt, with ZC showing small fluctuations at 1.5H and both ZY and TYC showing small fluctuations at 1H, and both reaching a minimum at 7H. The rate of change in dMWD with distance from the shelterbelts showed an increasing trend with increasing soil depth for all four shelterbelt types (in the 0–10 cm soil layers of YS, ZC, ZY, and TYC, dMWD progressed from 1.47 to 3.47, 1.88 to 4.59, 1.10 to 4.41 and 0.41 to 1.90 mm, respectively; the comparable values in the 10–20 cm soil layer were 1.90–3.84, 2.38–4.77, 2.28–4.35 and 0.61–1.64 mm, respectively; and in the 20–40 cm soil layer, 1.99–4.05, 3.05–4.83, 2.08–4.01 and 0.56–1.73 mm). However, wMWD showed a trend of increasing and then decreasing with increasing soil depth, and the 0–10 cm soil layer was significantly larger than the 20–40 cm layer. The variation of dGMD after shelterbelt in the 0–10 cm soil layers YS, ZC, ZY and TYC ranged from 0.61–2.83, 1.00–2.99, 0.54–2.87 and 0.25–0.79 mm, respectively; in the 10–20 cm soil layer 0.89–2.15, 1.68–3.48, 1.17–2.68 and 0.33–0.78 mm, respectively; and in the 20–40 cm soil layer 0.98–2.06, 1.81–3.25, 1.03–2.37 and 0.41–0.82 mm, respectively, with ZC and TYC showing a gradual increase with increasing soil depth.

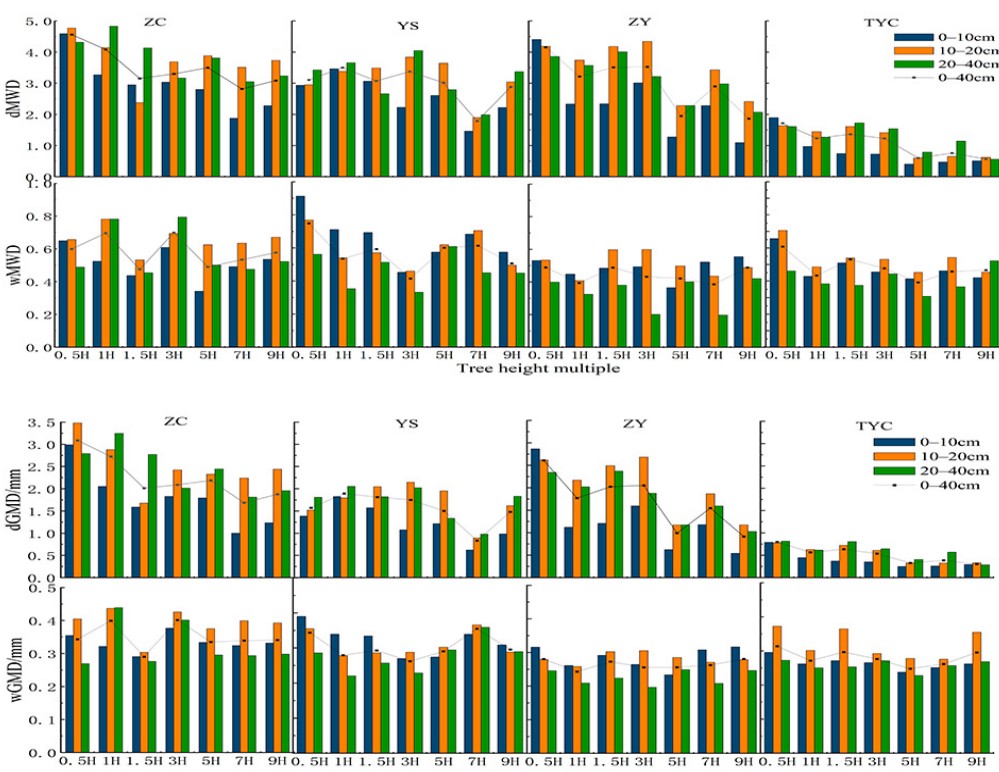

**Figure 7.** Change in the stability of soil aggregates (dMWD, dGMD, wMWD and wGMD) at different distances from shelterbelts.

The fractal dimension (D) of the soil dry and wet aggregates at different shelterbelts are shown in Figure 8. The mean values of dD in the transformed shelterbelts (YS, ZC, ZY) were smaller than those of degraded shelterbelt (TYC), and the transformation methods differed in the magnitude of dD values, with YS > ZY > ZC, while wD did not vary significantly among the transformed shelterbelts. The mean values of YS, ZC, ZY and TYC in the 0–40 cm soil layer were 2.46, 2.20, 2.42 and 2.51 mm, respectively, where ZC had no significant effect on dD with increasing distance from the shelterbelt and YS and ZY showed an increasing trend of dD with increasing distance, while the TYC shelterbelt had smaller dD and wD values in 0.5H–1H, then jumped to a much higher level at 1.5H and beyond. From the vertical variation of soil in the YS, ZY and TYC shelterbelts, dD shows an increasing trend with increasing soil depth, with 0–10 cm soil layers greater than 20–40 cm, while in the ZC shelterbelt, dD does not change significantly with increasing soil depth.

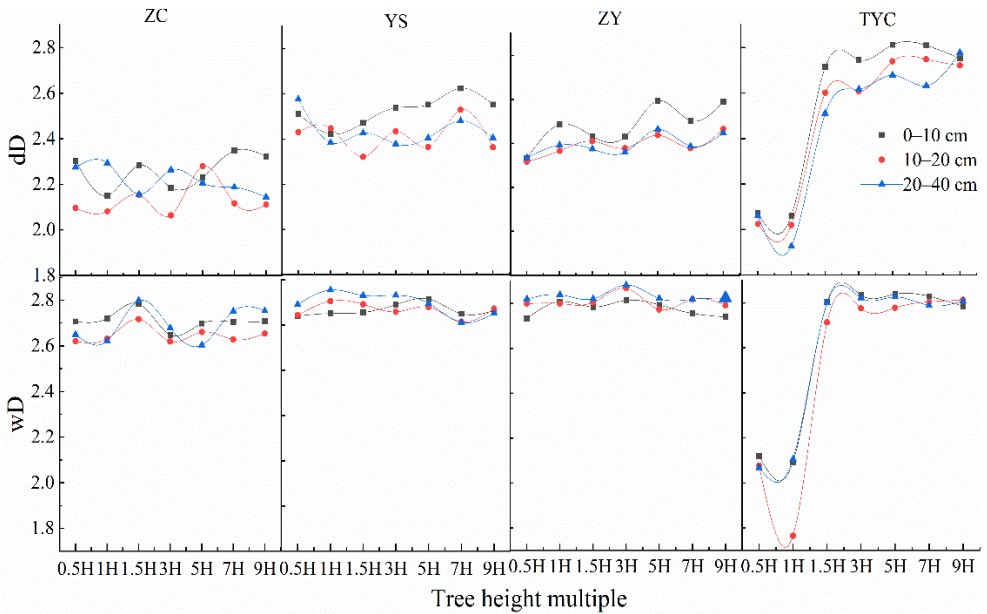

**Figure 8.** Change in soil fractal dimensions dD and wD at different distances from shelterbelts.

The results of the study on the stability of dry and wet sieve aggregates show that, compared with degraded shelterbelts, the transformation of farmland shelterbelts can significantly improve the stability of the aggregates and promote healthy soil structure. We see differences in the stability of the aggregates depending on the shelterbelt transformation method, with the best effect in ZC shelterbelts. The distance between the shelterbelts also had an effect on the stability of the aggregates: the dry sieve stability showed a trend of higher aggregates stability in samples taken closer to the shelterbelts, decreasing with increasing distance, while the wet sieve did not change significantly, and the dry sieve aggregates stability was significantly higher than that of the wet sieve.

### 3.3. Effect of Shelterbelts Transformation on Soil Erodibility

The change in soil erodibility K values from different shelterbelts are shown in Figure 9. Overall, the mean K values of each soil layer are smaller in the transformed shelterbelts (YS, ZC and ZY) than in the degraded shelterbelt (TYC), and there are differences in the magnitude of the K values depending on the transformation method, with ZY > YS > ZC. In the 0–40 cm soil layer, the mean K values for the YS, ZC, ZY and TYC shelterbelts were 0.029, 0.023, 0.030 and 0.083, respectively. In the 0–10 cm soil layer, the K values for each shelterbelt showed a gradual increase with increasing distance from the shelterbelt, with TYC showing a greater variation in K values than YS, ZC, and ZY. There was less variation and a more consistent trend among YS, ZC, and ZY in the 10–20 cm and 20–40 cm soil layers, while TYC showed a smaller K value at 0.5H–3H, then, after a significant increase at

3H, tended to level off. The range of K values after shelterbelts in the 0–10 cm soil layers YS, ZC, ZY and TYC was 0.022–0.059, 0.018–0.039, 0.018–0.067 and 0.051–0.134, respectively. In the 20–40 cm soil layer, the ranges were 0.019–0.045, 0.017–0.028, 0.019–0.038 and 0.046–0.119 mm, respectively, showing a decreasing trend with increasing in soil depth, consistently lower than the values in the 0–10 cm soil layer. This indicates that the transformed shelterbelts are associated with superior erosion resistance of farmland soils, though this gradually decreases with increasing distance from the shelterbelt.

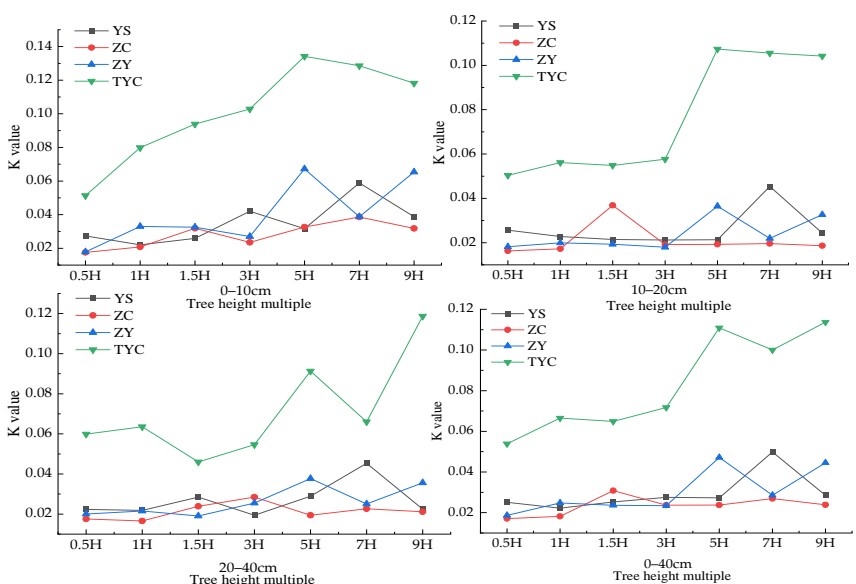

**Figure 9.** Change in soil erodibility K values at different distances from shelterbelts.

*3.4. Comprehensive Evaluation of Shelterbelts Transformation on the Stability of Soil Aggregates*

Soil aggregate stability evaluation index values (F) are shown in Figure 10. The results showed that in the 0–10, 10–20 and 20–40 cm soil layers, YS shelterbelt mean F values were 4.39%, 4.41%, and 2.57% respectively, which were 372.47%, 427.66%, and 211.73% higher than TYC. ZC mean values were 9.46%, 8.20% and 5.05% respectively, which were 916.79%, 882.04%, and 319.55% higher than TYC. ZY mean values were 4.35%, 3.71%, and 2.05% respectively, which were 67.83%, 343.88%, and 189.04% higher than TYC respectively. The method of shelterbelt transformation has a significant effect on the size of F, with ZC > YS > ZY. In addition, the distance from shelterbelt has a significant influence on the stability of aggregates; for example, YS and ZY in the 0–40 cm soil layer show a "W"-shaped variation in F with increased distance from shelterbelt, decreasing at 1H and 1.5H, while ZC shows a gradual decrease with distance from shelterbelt, and TYC shows a positive value in the distance of 0.5H–1.5H, then a negative value after 1.5H, with a decreasing trend with increased distance from shelterbelt. The effective protection range of ZC is 1.5H, YS and ZY is 1H, and TYC is 0.5H. The comprehensive evaluation of F indicates that shelterbelt transformation is conducive to improving the stability of soil aggregates within the shelterbelt's protective scope, and that different transformation methods have different effects on their improvement, with ZC having the best effect.

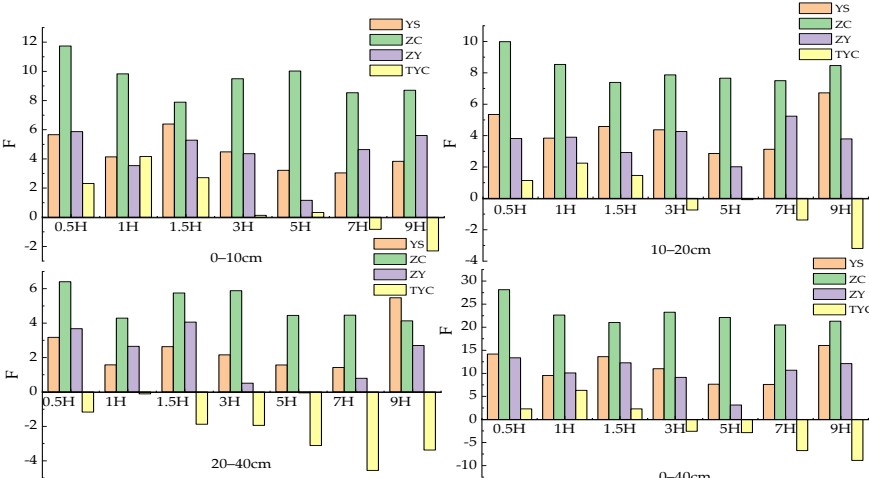

**Figure 10.** Comprehensive evaluation index of soil aggregates stability at different distances from shelterbelts.

*3.5. Connection between Stability of Aggregates and Soil Properties*

The relationships among soil aggregate stability, K values, and soil properties were analyzed using the Pearson correlation method (see Table 2). The SOC and TN showed highly significant positive correlations with aggregates wMWD and wGMD, positive correlations with aggregates dMWD and dGMD, and negative correlations with wD, dD and K. Clay and silt content showed highly significant positive correlations with aggregates dMWD and dGMD, positive correlations with aggregates wMWD, wGMD and wD, and highly significant negative correlations with dD and K values. Sand content was highly significantly positively correlated with aggregate dD and K values, highly significantly negatively correlated with aggregate dMWD, dGMD and wGMD, and negatively correlated with aggregate wD and wMWD. This means that the lower the sand content and bulk density of the soil, the higher the organic matter content, the lower the D value, the higher the MWD and GMD values, and the higher the stability of the aggregates.

**Table 2.** Pearson correlations between the stability of soil aggregates, K values and soil properties.

|  | dD | wD | dMWD | wMWD | dGMD | wGMD | K | BD | Clay | Silt | Sand | SOC | TN |
|---|---|---|---|---|---|---|---|---|---|---|---|---|---|
| D | 1 | | | | | | | | | | | | |
| wD | 0.835 ** | 1 | | | | | | | | | | | |
| dMWD | −0.368 ** | 0.080 | 1 | | | | | | | | | | |
| wMWD | −0.139 * | −0.167 * | 0.196 ** | 1 | | | | | | | | | |
| dGMD | −0.351 ** | 0.092 | 0.956 ** | 0.236 ** | 1 | | | | | | | | |
| wGMD | −0.226 ** | −0.210 ** | 0.201 ** | 0.887 ** | 0.274 ** | 1 | | | | | | | |
| K | 0.426 ** | −0.038 | −0.866 ** | −0.172 ** | −0.751 ** | −0.197 ** | 1 | | | | | | |
| BD | 0.093 | −0.027 | −0.011 | −0.025 | −0.044 | −0.090 | 0.045 | 1 | | | | | |
| Clay | −0.292 ** | 0.034 | 0.457 ** | 0.113 | 0.453 ** | 0.282 ** | −0.525 ** | −0.327 ** | 1 | | | | |
| Silt | −0.304 ** | 0.047 | 0.511 ** | −0.001 | 0.542 ** | 0.139 * | −0.494 ** | −0.258 ** | 0.358 ** | 1 | | | |
| Sand | 0.364 ** | −0.050 | −0.591 ** | −0.066 | −0.608 ** | −0.254 ** | 0.618 ** | 0.356 ** | −0.817 ** | −0.831 ** | 1 | | |
| SOC | −0.075 | −0.102 | 0.004 | 0.339 ** | 0.024 | 0.326 ** | −0.033 | −0.248 ** | 0.210 ** | −0.059 | −0.088 | 1 | |
| TN | −0.142 * | −0.102 | 0.070 | 0.290 ** | 0.115 | 0.364 ** | −0.080 | −0.286 ** | 0.413 ** | 0.009 | −0.251 ** | 0.665 ** | 1 |

Notes: * means $p < 0.05$, and ** means $p < 0.01$.

### 3.6. Shelterbelt Characteristics, Soil Texture and Soil Properties on the Variation of Soil Aggregate Characteristics at Different Distances from Shelterbelts

As shown in Table 3, overall, porosity provides the greatest explanation for changes in the stability and erodibility of soil aggregates. Porosity is the most influential parameter in each soil layer, providing the highest percentage of interpretation. For example, in the 0–10 cm, 10–20 cm and 20–40 cm soil layers, porosity accounted for 27.6%, 49.6%, and 35.3% of the variation in soil aggregate stability and erodibility induced by shelterbelts, respectively (Table 3). Silt explained more variation in the deeper soils compared with the surface layer. Soil in the 0–10 cm layer showed no significant explanation for variation, while silt accounted for 0.3–1.2% of the variation in the 10–20 and 20–40 cm layers. The distance from the shelterbelt had the second highest explanatory power for changes in soil aggregate stability and erodibility, showing a significant percentage ($p < 0.05$) in all soil layers though exhibiting a gradual decrease with increasing soil depth. In addition, soil sand content and total nitrogen content significantly explained the variation in soil aggregate stability and erodibility associated with shelterbelts, the explanatory power of which ranged from 1.2–5.8% and 0.2–2.2% for different soil layers.

**Table 3.** Comparison of the explanatory power of shelterbelt characteristics, soil texture and soil properties in the variation of soil aggregate characteristics in farmland at different distances from shelterbelts from RDA ordination-related conditional term effects, excluding co-linear effects.

| Soil Depth | Name | % Explained | Pseudo-F | $p$ | RDA Ordination Figure |
|---|---|---|---|---|---|
| 0–10 cm | Porosity | 27.6 | 27.4 | 0.002 | |
| | DIS | 10.7 | 12.3 | 0.002 | |
| | Sand | 1.5 | 1.8 | 0.190 | |
| | BD | 1.0 | 1.1 | 0.302 | |
| | SOC | 0.4 | 0.5 | 0.650 | |
| | Height | 0.4 | 0.5 | 0.660 | |
| | Clay | 0.5 | 0.6 | 0.604 | |
| | TN | 0.2 | 0.2 | 0.838 | |
| 10–20 cm | Porosity | 49.6 | 68.0 | 0.002 | |
| | DIS | 7.0 | 11.0 | 0.002 | |
| | TN | 2.2 | 3.6 | 0.03 | |
| | Height | 1.1 | 1.8 | 0.14 | |
| | BD | 0.6 | 1.0 | 0.362 | |
| | Clay | 0.7 | 1.1 | 0.344 | |
| | Silt | 0.3 | 0.4 | 0.648 | |
| | SOC | 0.2 | 0.3 | 0.798 | |
| 20–40 cm | Porosity | 35.3 | 35.4 | 0.002 | |
| | DIS | 6.8 | 10.1 | 0.002 | |
| | Sand | 2.8 | 3.3 | 0.036 | |
| | TN | 1.8 | 2.2 | 0.096 | |
| | Silt | 1.2 | 1.4 | 0.226 | |
| | Height | 0.6 | 0.7 | 0.554 | |
| | SOC | 0.6 | 0.7 | 0.490 | |
| | BD | 0.2 | 0.3 | 0.832 | |

## 4. Discussion

### 4.1. Effect of Shelterbelt Transformation on Particle Size Distribution of Soil Aggregates

In this study, the transformation of the farmland shelterbelts led to a significant improvement on farmland soil structure. This is because the transformation of shelterbelts has resulted in a significantly higher proportion of dry and wet macroaggregates compared to the conditions around degraded shelterbelts. We found that dry sieves in transformed shelterbelt-protected farmland (YS, ZC, ZY) were mainly dominated by >5 mm and 2–5 mm grain size aggregates, while degraded shelterbelt-protected farmland (TYC) were mainly dominated by 0.25–0.5 mm and <0.25 mm grain size aggregates. It shows that the transformation of farmland shelterbelts can significantly change the grain size distribution of soil dry aggregates compared to degraded shelterbelts, causing the size of aggregates to change from small to large particle size.

These studies confirm similar results previously reported [43,44]. The reason for this may be that degraded shelterbelts have a significantly lower ameliorating effect on the farmland soil due to the degradation of the shelterbelt structure (thus reducing its ability to mediate wind speed), resulting in a lower proportion of large aggregates. Shelterbelt wet sieves found mainly 0.25–0.5 mm and <0.25 mm size aggregates, with relatively stable variation in 0.25–0.5 mm size aggregates compared to >5 mm aggregates. This may be due to the fact that when soils are exposed to water or wind erosion, >5 mm particle size aggregates are more easily decomposed, affecting the formation of water-stable macroaggregates [45]. In addition, the effect of shelterbelt transformation on the particle size of dry aggregates was found to be significantly greater than that of wet aggregates. This may be due to the semi-arid climate type of the region and the sandy type of the soil texture. Due to its loose soil texture and mostly wind eroded environment, the soil easily breaks up under the action of dry sieves. As a result, we find a significantly greater effect on dry aggregates than on wet aggregates.

Farmland soil aggregate particle size distribution is influenced by distance from the shelterbelt. For example, the ZC shelterbelt has a higher content of macroaggregates (>0.25 mm) than other shelterbelts, and the content level of these macroaggregates is higher closer to the shelterbelts, fluctuating with increasing distance. The effect of distance from shelterbelts on the particle size distribution of dry sieved aggregate is significantly higher than that of wet sieves. This may be due to the fact that the farmland is affected by the microclimate—specifically, the soil hydrothermal environment created by the shelterbelts—which inevitably affects the content of organic matter material in the farmland soil [18]. This explains the variation of aggregate content with distance from shelterbelts in the current study.

Our results clearly show that the transformation of shelterbelts can significantly improve the particle size distribution of soil aggregates, inducing a shift from small to large sized aggregates, and that the transformation methods and distances from shelterbelts affect the size distribution of aggregates and soil structure, providing possible pathways for improving farmland soil.

### 4.2. Soil Aggregates Stability in Reaction to Shelterbelts Transformation

The MWD and GMD of the soil are key indicators of the stability of the aggregates: the higher the MWD and GMD values, the more stable the aggregates and the higher the degree of aggregation [46,47]. The fractal dimension (D) represents the geometry of the soil structure and reflects the size of the soil particle size distribution: the smaller the D value, the better the soil stability [48]. The results show that the dMWD and dGMD of farmland soil aggregates are higher and dD is lower in farmlands protected by transformed shelterbelts (YS, ZY, and especially ZC) compared with those protected by degraded shelterbelts (TYC). This suggests that the structure of farmland soil has become more stable after the shelterbelt transformations, and the stability of aggregates has increased significantly. This is similar to the findings of Wang, et al. [49,50], who found that afforestation in central China promotes the formation of large aggregates and improves the stability of soil aggregates to promote

the physical conservation of soil organic carbon. This may be attributed to the clay and silt content of farm soils and the periodic wet and dry variation (wD) [51].

In the current study, the transformation of degraded shelterbelts increased the clay and silt content of farmland soils (Figure 4), promoting wD and leading to the formation of large aggregates. In addition to cultivation activities, planting shelterbelts on farmland can spur offsetting processes that have a strong influence on the stability of aggregates, one way or another [1,52]. Here, the clay and silt content of the soil, the cultivation activity, and the wet and dry cycles are factors affecting the stability of the aggregates [53]. These factors, both positive and negative, complement each other in the process of aggregation or declustering.

As shown in Table 2, the correlations of dMWD and dGMD with sticky and powdery particles indicates that clay and silt content have an important influence on the stability of dry aggregates. Compared with the lower wMWD and wGMD, the higher dMWD and dGMD are mainly associated with the higher amounts of large particle aggregates obtained through dry sieving. This is consistent with the findings of Sainju [54] in the United States, where researchers reported higher dMWD values than wMWD values in semi-arid areas of eastern Montana and western North Dakota, attributed to the semi-arid climatic conditions and the loose soil texture type of the area. Wet sieving tends to destroy soil particles in large aggregates, especially in semi-arid regions.

In addition, in this study it was found that the transformation method significantly influenced the stability of dry aggregates on the farmland, showing that the dMWD and dGMD were significantly higher on the ZC protected farmland than on the YS, followed by the ZY. Mean values of dMWD and dGMD decreased with increasing distance from shelterbelts and increased with soil depth. However, the effect on the stability of wet sieve aggregates was not significant and wMWD and wGMD varied less with distance from shelterbelts, though they did increase with soil depth.

Based on the fact that dMWD and dGMD show highly significant positive correlations with clay and silt content and highly significant negative correlations with sand content, the mechanical composition parameters can be used to predict the sensitivity of the farmland soils to wind erosion in the semi-arid areas of the black soil region of western Heilongjiang Province [55]. However, given that wMWD and wGMD are only weakly positively correlated with clay and silt content and weakly negatively correlated with sand content (Table 2), in terms of soil aggregate stability, dry sieving is a good alternative to wet sieving, especially in semi-arid wind eroded areas. Meanwhile, dMWD and dGMD can be used as important parameters for modeling and predicting the sensitivity of the farmland soil to wind erosion in semi-arid environments.

The positive correlation between dD and wD (R = 0.835, Table 2) further illustrates the significant relationship between these two indicators in our study: wD explains 83.5% of the variation and dD explains the same degree of variation. Based on principal component and correlation analyses, we find that the simpler and faster dry sieving method for aggregates can be used to replace the complex and tedious wet sieving method in the semi-arid soils of the Black Soil Plain in western Heilongjiang Province [56]. However, we note that stability indices that are poorly predicted in the soil types of our study may be well predicted in other soil types and climatic environments. Thus, further research is needed on different soil types and climate conditions.

*4.3. Soil Erodibility K Values in Response to Shelterbelts Transformation*

The soil erodibility factor, K, reflects the stability of the physical structure of the soil and is closely related to the stability of the soil aggregates [57]. The lower the K value, the greater the resistance to erosion. In this study, we found that the K values of farmland soils protected by transformation shelterbelts were significantly lower than those protected by degraded forest belts in the 0–10 cm, 10–20 cm and 20–40 cm soil layers. The K-values vary according to the mode of transformation, with ZC > YS > ZY, and gradually increase with distance from the shelterbelts while decreasing with soil depth.

This indicates that the transformation of farmland shelterbelts helps to improve the erosion resistance of farmland soils within a certain distance from the shelterbelts, and that all of the transformation modes improve the erosion resistance of farmland. The distance from shelterbelts affects the soils' resistance to erosion, particularly the topsoil. The transformed shelterbelts effectively improve the physical and chemical properties of the farmland soils within a certain distance, promoting the formation of new aggregates and thus improving erosion resistance. This is consistent with the results of Erktan, et al. [58] who found that vegetation can reduce the susceptibility of soils to erosion by increasing the stability of soil aggregates. As the stability of the soil aggregates increases, we find that this enhances the resistance of the soil to erosion, consistent with previous studies [59].

*4.4. Effect of Shelterbelts Structure on Soil Physicochemical Properties, Aggregate Stability, and Erodibility*

Different transformation modes result in shelterbelts with different structural characteristics. This may lead to differences in microclimatic characteristics within the shelterbelt-protected farmland, ultimately affecting the mechanical composition and physical and chemical properties of the farmland soils as well as the stability of soil aggregates and soil erodibility within certain distances from shelterbelts [60,61]. In the present study, it was found that the degree of porosity of shelterbelts was the most important factor influencing the physical and chemical properties of the soil, the stability of the aggregates, and the erodibility of the soil ($p < 0.001$, Table 3). However, the most important factor influencing the development of soil aggregates is the mode of shelterbelt transformation [62]. For example, compared to degraded shelterbelts in this study, transformed shelterbelts significantly improved soil aggregate stability, soil C and N, and reduced soil erodibility (Figure 4), and the effects on soil aggregates stability and soil erodibility depended on the transformation method. These observations are also supported by previous studies on yellow soil regions of the the Loess Plateau in central China [8,62], where revegetation was effective in improving soil aggregates.

However, the effects of farmland shelterbelts on the stability of soil aggregates, erodibility, soil carbon, and nitrogen improvement all tend to decrease gradually with increased distance from the shelterbelts. The porosity of the shelterbelt is the main shelterbelt structure parameter driving changes in soil aggregate stability, soil erodibility, and soil properties, with effects decreasing with increased porosity. We also found small fluctuations in soil physicochemical properties, cluster stability, and soil erodibility as the distance from the shelterbelts increased. In particular, transformed shelterbelts show greater effects than do degraded shelterbelts, and dry aggregates are more significant than wet aggregates (Figure 7). This phenomenon may be due to the porosity of the shelterbelts, which determines the form of wind penetration through or over the shelterbelts, resulting in differences in the intensity and condition of the wind distribution behind the shelterbelts, affecting the strength of wind erosion on the farmland soils, and causing fluctuations in soil stability.

## 5. Conclusions

In this study, we examined farmland shelterbelts that have been transformed to reduce the occurrence of wind erosion, increase the content of clay and silt content, and improve the distribution and stability of soil aggregates and erosion resistance of farmland. Our combined analysis of the soil aggregate stability evaluation index values (F) further confirms this result, showing that the mode of shelterbelt transformation matters—specifically, ZC > YS > ZY. In addition, overall protection decreases with increasing distance from the shelterbelts, especially in the topsoil, where ZC provides the best protection.

We also found that the effect of farmland shelterbelts on dry aggregates was significantly higher than that of wet aggregates. dMWD and dGMD can be used as important parameters for modeling and predicting the sensitivity of agricultural soils to wind erosion in semi-arid environments. Dry screening has the same effect as wet screening on the distri-

bution and stability of aggregates in this region, and wind erosion is the main type of soil erosion in semi-arid arid regions with limited rainfall, compared to wet and semi-humid areas. In addition, the variations in soil aggregation stability, soil erodibility and soil properties are mainly influenced by shelterbelt porosity. The stability and resistance to erosion of farmland aggregates in the *Pinus sylvestris var. mongolica* shelterbelt is significantly better than in the mixed *Populus × xiaohei-Pinus sylvestris var. mongolica* and *Plicae asperata* shelterbelt. These results benefit the post-transformation assessment of farmland shelterbelts and provide important and novel insights into how the stability characteristics of farmland soil aggregates, soil erodibility, and structural features of shelterbelts interact at the scale of shelterbelt networks.

**Author Contributions:** Conceptualization, T.K. and B.L.; methodology, T.K.; software, T.K.; validation, B.L., L.W. and T.K.; formal analysis, T.K.; investigation, T.K., S.W., Y.S., W.Z. and G.W.; data curation, T.K.; writing—original draft preparation, T.K.; writing—review and editing, B.L. and M.H.; funding acquisition, B.L. All authors have read and agreed to the published version of the manuscript.

**Funding:** This research was funded by the Applied Technology Research and Development Program of Heilongjiang Province, China (GA20B401), the National Key Research and Development Program of the 14th Five-Year Plan of China (2021YFD1500705) and National Natural Science Foundation of China (41877416).

**Institutional Review Board Statement:** Not applicable.

**Data Availability Statement:** Not applicable.

**Acknowledgments:** We gratefully acknowledge the Applied Technology Research and Development Program of Heilongjiang Province, China (GA20B401), National Key Research and Development Program of China's 14th Five-Year Plan (2021YFD1500705), National Natural Science Foundation of China (41877416) for funding this work.

**Conflicts of Interest:** The authors declare no conflict of interest.

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
