# Peer review of "Effects of Shelterbelt Transformation on Soil Aggregates Characterization and Erodibility in China Black Soil Farmland"

_agriculture, doi:10.3390/agriculture12111917_

Round 1

Reviewer 1 Report

In this work, the authors evaluate the impact of shelterbelt transformation on the stability of farmland soil aggregates and soil erodibility. They used a theoretical approach. However, the present manuscript, many of the presented trends are exposed with quite a lack of data interpretation. The manuscript requires minor revisions before being considered for publication in “agriculture” or any journal in this field.

1. Abstract: Add quantitative data and its interpretation

2. Pictures 1, 2 and 3 are a blur. Please fix it!

3. Researchers generally present the analysis results qualitatively, even though the theoretical approach is carried out. It is advisable to provide quantitative data, such as K [line 228] or F [line 246] and make the proper interpretation for them

Author Response

Dear Reviewers:

Thank you for your letter. Attached please find our revised manuscript “Effects of shelterbelt transformation on soil aggregates characterization and erodibility in China black soil farmland” (1978678) and our point-by-point response to the comments of the reviewer and the editor. We thank the reviewers for the time and effort that they have put into reviewing the previous version of the manuscript. Their suggestions have enabled us to improve our work. We carefully modified the paper in response to the comments and suggestions. All coauthors have carefully checked and improved the contents of the manuscript and have agreed to submit the revised manuscript. Appended to this letter is our point-by-point response to the comments raised by the reviewers. We have highlighted the changes by using the track changes mode in Microsoft Word: revisions in the text are shown using yellow highlight for additions, and strikethrough font for deletions. The responses to the reviewer's comments are marked in red and presented below. We hope that the revised manuscript can be accepted for publication in the Journal of Agriculture.

Sincerely,

Tongwei Kong

Reviewer 2 Report

The study has provided some important outlines for a better understanding of shelterbelts around farmland  (one of the component of phytomelioration) which are very effective in arid and semi-arid areas, mostly because of reduce wind speeds, control soil erosion, and improve microclimates. Shelterbelts can be also an effective measure to stabilize crop production and also promoting ecological landscape functions.  The main objectives of the study were, to assess and compare the effects of different shelterbelt transformation methods on soil aggregates and soil erodibility, and to explore the relationship between soil aggregate characteristics and soil erodibility with soil physicochemical properties and the shelterbelts’ structural characteristics. Authors use the study site which was located in Gannan County, in the northwestern part of the Songnen Plain.   The Authors have shown very interesting results. Among the many one of the important is that the stability and resistance to erosion of farmland aggregates in the Pinus Sylvestris var. mongolica shelterbelt was significantly better than in the mixed Populus × xiaohei – Pinus sylvestris var. mongolica and Picea asperata shelterbelt.  Work written correctly,  research methods selected correctly, results presented in a clear manner, enough literature included, although it was found the bad citation in a few cases. All needed corrections are marked in the text of the paper.

After all corrections the paper can be publish at the MDPI Agriculture Journal

Author Response

(The authors gave the same response as above.)

Reviewer 3 Report

The study has provided good results, but there are some flaws that need to be fixed.

I have provided the necessary explanations in the text.

Author Response

(The authors gave the same response as above.)

Round 2

Reviewer 3 Report

Necessary corrections have been made.

The article is now printable.

Regards